# CE Marking of Construction Products—Evolution of the European Approach to Harmonisation of Construction Products in the Light of Environmental Sustainability Aspects

Sebastian Wall

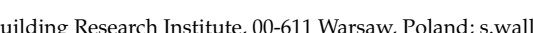

Building Research Institute, 00-611 Warsaw, Poland; s.wall@itb.pl

**Abstract:** European harmonisation of construction products provides a uniform expression of performance aspects relevant for essential characteristics coming from notified technical building regulations of the EU Member States. Since the current regulation has been in force for over seven years, this study evaluates further possibilities of its evolution, including a more efficient approach to implementing environmental sustainability aspects. The provided research is based on qualitative analysis of the past and current legislation, official documents, related guidance, judgements, scientific articles and the author's practical experience coming from participation in the European committees, organisations and standardisation activities. Various legislative techniques and regulatory tools that could be potentially used to review the Construction Products Regulation are analysed and compared with regards to their impact on the inclusion of environmental sustainability principles. Therefore, the objective of this research is to provide substantive grounds that can be directly or indirectly used in the policymaking processes on the European and national level.

**Keywords:** CPR; 305/2011; CE marking; construction products; environmental sustainability

## 1. Introduction

According to the data presented by the European Commission (EC), the construction sector is relevant for an impressive number of 18 million jobs and about 9% of GDP in the European Union [1]. The sector also contributes strongly to the environmental impact of the EU as European building stock is responsible for 40% of final energy consumption and 36% of all greenhouse emissions [2]. It is also perceived as an innovation-driven [3], highly professional area that should be a subject of effective legislative support. However, one of the biggest challenges, especially when the economic values need to be considered, has always been to ensure that the constructor sector contributes properly to one of the most important principles of the EU, which is the free trade of goods. Sector relevant goods are mainly construction products—the goods manufactured to order to be incorporated in construction work, in a permanent manner, which performance has an influence on the level how the construction work contributes to the level of characteristics that are required by national laws (for example building technical regulations). This distinction between product's performance and the technical conditions to be fulfilled by the construction works also constitutes the borderline between the roles and responsibilities of economic operators, the Union and the EU Member States (MS), as a background for the still-living concept of the European harmonisation of construction products that was introduced some 30 years ago.

Before the end of the 80s, the manufacturers placing construction products in the national markets of EU Member States were confronted with various, not mutually recognised national systems, approval procedures, certifications or markings. That obviously caused increased cost for cross border trade and use of the product, due to different administrative barriers. Therefore, in the White Paper on Completing the Internal Market of 1985 [4], the

European Commission announced the plan to establish harmonised code for buildings to achieve comparability and a single market for products and building components.

Consequently, in February 1989, the Directive 89/106/EEC (so-called Construction Products Directive—CPD) [5] was published in the Official Journal of the European Union (OJEU) as the first European piece of legislation that had been developed to harmonise the area of construction products. The objective of the CPD was ambitious, as it was intended to approximate all national provisions and procedures related to the performance and use of construction products in the whole EU area. However, this approximation was clearly not affecting the field of technical regulations for construction works, especially regarding citizens' safety in the sole competencies of national authorities of the Member States [6].

The intention of the European legislator was from the beginning to achieve the objectives via the establishment of harmonised technical specifications, attestation of conformity (AoC) procedures, including special roles for notified bodies, approval bodies, and finally, CE marking of construction products. The CE marking and the declaration of conformity (DoC) were introduced as the key tools of the directive and as publicly available proof of conformity of a particular product with the harmonised specification. As a directive, the CPD formed a part of the European law that was transposed by national regulations. It needs to be stressed that due to the principle of conformity with harmonised specifications, but not with the EU directive itself, the CPD was somehow different from "New Approach" directives [7]. As mentioned above, this was a clear repercussion of the distribution of competencies between MSs and the EC.

The directive in Annex I referred to six categories of general requirements to be fulfilled by construction works, so-called Essential Requirements (ERs). The structure of requirements reflected the nature and content of works-related regulatory provisions of the Member States. Therefore, the ERs were referring to:

− Mechanical resistance and stability (ER 1);
− Safety in the case of fire (ER 2);
− Hygiene, health and the environment (ER 3);
− Safety in use (ER 4);
− Protection against noise (ER 5);
− Energy economy and heat retention (ER 6).

The ER 3, potentially covering selected aspects of protection of people or natural resources, reflected the regulatory state of the art in the MSs at the beginning of the 90s. Therefore, it was practically limited, referring to very few environmental impacts of construction works. The ER 3 was, to a certain extent, linked to the presence or emission of dangerous substances in construction products and environmental impacts that could occur during the use phase of construction works. Aspects related to the environmental impacts of manufacturing products, construction processes, demolition or end-of-life phase were clearly not considered. Similarly, ER 6 was concentrated only on the energy use on ventilation, heating and air-conditioning only during construction work.

According to art. 3 of the CPD, the Essential Requirements and the interpretative documents containing necessary explications (e.g., regarding covered products, characteristics and intended uses) formed a basis for further technical harmonised specifications. The system of technical specifications focused mainly on widely available (practically via national standardisation bodies) harmonised European standards (hEN), developed based on standardisation mandates addressed by the Commission to the European Committee on Standardisation (CEN). The mandates were issued in the case when the Member States, the industry and the Commission jointly agreed that the trading of a certain category of construction product is affected by legal requirement related to its performance, which results from notified national technical regulation, that finally might be assessed as a barrier to trade. Similar mandates were issued to develop Guidelines for European Technical Approvals (ETAG) for innovative, non-standardised products, for which European Technical Approval (ETA) could be issued by approval body, member of the European Organisation for Technical Approvals established in 1990 (EOTA), as an assessment of fitness for

use. However, for individual products, ETAs were also issued based on Common Understanding on the Assessment Procedure (CUAP) in accordance with art. 9(2) of the CPD, without mandating process and relevant guidelines. To a certain extent, such a concept reflected both the needs of "well known" standardised products and non-standardised innovation, which required a slightly different, more individual approach. However, it needs to be noted that this approach always required at least consent from the Commission that influenced in this way time to the market of certain products.

Finally, for a product in conformity with a harmonised standard or ETA, the CE marking was possible (see Figure 1), and the obligation to CE mark could result only from a national transposition of the CPD.

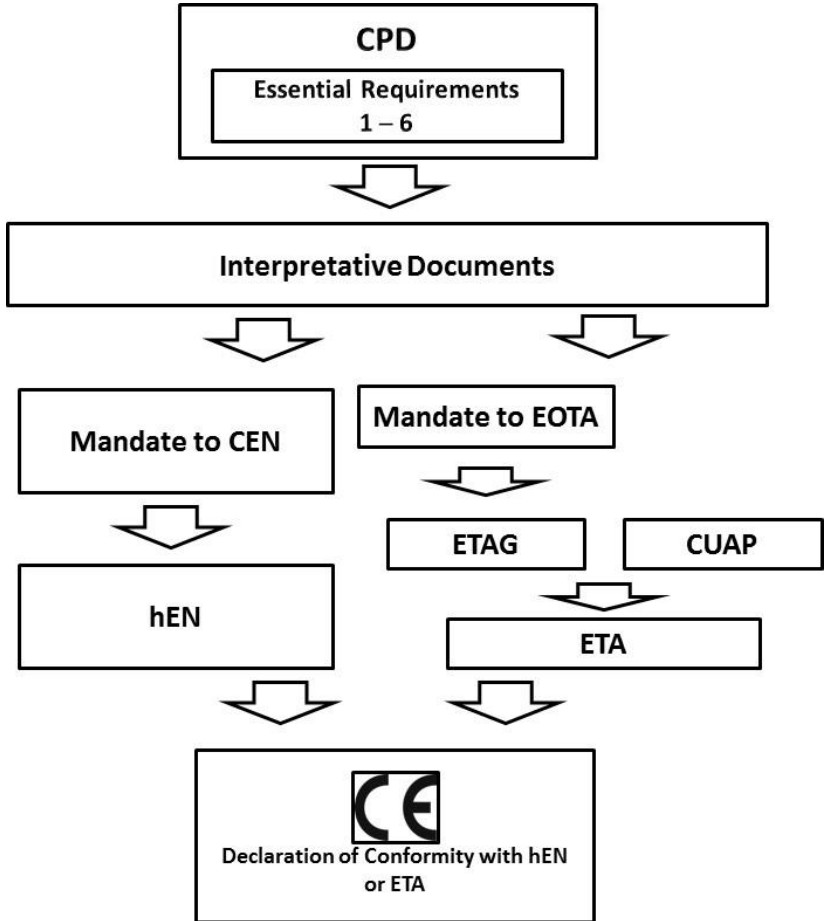

**Figure 1.** From the Essential Requirements to the CE marking of construction product.

The assessment procedures under the CPD used a set of attestation of conformity systems that considered different tasks for a manufacturer and the third parties (notified laboratory or notified certification body). Appropriate AoC system (1+, 1, 2+, 2, 3, 4) was indicated in relevant EC Decisions, developed based on an evaluation of risk related to the role of a particular product in construction works in relation to essential requirements, the variability and constancy of product characteristics and tasks within the manufacturing process. The most onerous one (1+) required full product certification by the notified body when the least onerous one (4) was fully dependent on the manufacturer's activities.

Specific issues related to the practical execution and transposition of the directive were clarified in other interpretative and guidance documents elaborated jointly by the Commission and the Member States. The interpretative documents referred to several areas of common interest, among others, transposition of specific ERs into essential characteristics of construction products [8]. The papers were widely used, transposed and translated in

the whole EU for general use by manufacturers and authorities, also in the fields outside the scope of the CPD.

In relation to dangerous substances, individual Guidance Paper H [9] was addressed to writers of harmonised technical specifications to achieve coherence with the mandate(s). Specific terms and definitions were presented to support the development of hENs and ETAGs that should take into account in a proper way national or European provisions, regulated substances and related products. This means that legal requirements used in the directive and the mandate were translated into more comprehensive descriptions of the product's characteristics that could be understood and interpreted by the standardisers (technical people in most of the cases). Such explanation, together with information on the approach taken (when and how a certain substance has to be taken into account in the development process) and relevant examples complemented the legal text of the CPD.

In the middle of the first decade of the 21st century, the Commission decided to start the review process of the CPD. The conclusions of the study [10] showed a clear preference for a comprehensive revision of the directive that should aim at necessary clarifications, simplifications and stronger Commission control over harmonisation tools. At the same moment, some study participants indicated a preference for change of the future legislative form from directive to regulation that was perceived as more effective, due to potentially easier implementation and direct application without the need for long-term national transposition.

As a result of those considerations, in 2008, the Commission presented the first draft of a new regulation [11] which, after several changes that took place during the negotiation process in the European Parliament and the Council, was finally published in April 2011 as the EU Regulation No 305/2011, so-called Construction Products Regulation (CPR) [12]. The CPR came into force on 1 July 2013. The new approach to legislative tool also brought new expectations regarding the implementation of environmental or even sustainability principles.

The main objective of the new regulatory tool (CPR) is to ensure that information on construction product's performance is made available transparently, subsequently when the product is placed on the market. Such information is to be given in accordance with the common technical language expressed in harmonised technical specifications recognised in all EU Member States. Therefore, the principles of European harmonisation have been somehow downgraded from an ambitious level aimed at the content of national legislation to the level of technical information related to the product itself. Taking into account the distribution of the competencies between the Member States and the Commission, such change should theoretically improve implementing the EU law.

The regulation includes seven Basic Works Requirements (BWRs) that need to be considered a further evolution of ERs established under the CPD. The most important improvement came with extended requirements related to the environment and the use of resources [13].

Therefore, the content of ER3 evolved into BWR 3, referring to the whole life cycle of construction works, its impact on people, the environment and climate as a result of giving off toxic gases, emission and release of dangerous substances etc. Moreover, BWR 6 was extended to cover energy efficiency during construction, use and demolition of works.

However, the most important change relevant for the local and global environment is introducing a brand new BWR No 7 related to the sustainable use of natural resources [12].

"The construction works must be designed, built and demolished in such a way that the use of natural resources is sustainable and in particular ensure the following:

(a)     reuse or recyclability of the construction works, their materials and parts after demolition;
(b)     durability of the construction works;
(c)     use of environmentally compatible raw and secondary materials in the construction works".

The CPR authors also indicate a specific technical tool dedicated for the BWR 7 as Recital 56 says that "for the assessment of the sustainable use of resources and of the impact

of construction works on the environment Environmental Product Declarations should be used when available".

It needs to be underlined that the BWR7 was added to Annex I to the regulation to provide the Member States with the possibility to comprise future development of new laws on sustainable buildings or construction works. Therefore, in the case of the CPR, exclusively a new European requirement was created as a perspective tool, without the actual presence of national regulations notified in accordance with 98/34/EC Directive [14].

Additionally, to the BWR 3 and 7, in accordance with Art. 6(5) the information on the content of hazardous substances (REACH [15]) is to be provided with the construction products, when relevant. However, REACH information is not intended to be used further in the CPR data chain and is not classified as a performance of the product itself.

The CPR introduced a new tool that had not been used in European legislation before—a mandatory Declaration of Performance (DoP). In the CPR regime, the DoP is included as a key element in the information chain from the manufacturing plant to the end-user. According to art. 4 of the CPR, the DoP is issued when a construction product covered by the harmonised standard or European Technical Assessment (ETA) is placed on the market (Figure 2).

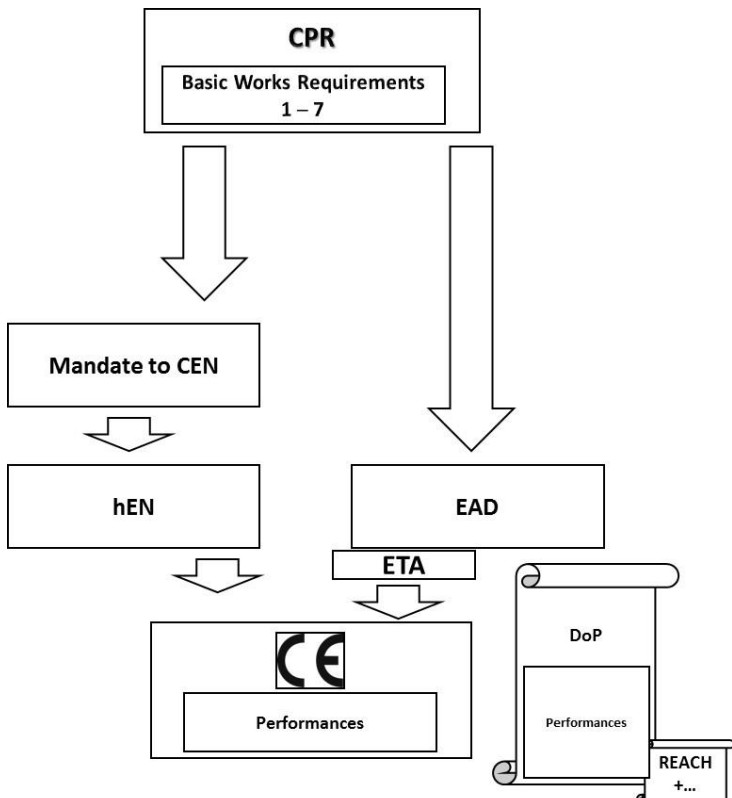

**Figure 2.** From BWRs to the DoP and CE marking of a construction product under the CPR.

A DoP is issued generally for the strictly identified product type, indicated by identification code and defined as a set of performances achieved with a certain production process and material composition of the product. It gives the information on performances of essential characteristics in relation to the intended use as indicated in the harmonised technical specification and on manufacturer details or on reference to the system of assessment and verification of constancy of performance (AVCP) used notified body involved.

The concept of DoP reflects the change of the paradigm from conformity assessment to an assessment of performance. AVCP systems are using modified philosophy of CPD AoC systems (system 2 was removed from the scheme as it had not been used anymore in any of the EC Decisions) to provide the basis for an assessment of product type further

control of stability of production process. Consequently, the third parties involved in the process are notified laboratories or notified certification bodies.

The CPR also changes the purposes of harmonised technical specifications, which are now used as tools for assessing and verifying the performance of construction products.

The first specification, which used is to be mandatory under the CPR, is still the harmonised European standard. A transitional assumption was made that standards cited under the CPD may be used for CPR-related purposes as the legacy of CPD was taken over. Even the footnotes in OJEU lists of citations were introduced, stating that in case of conflict between CPR and the standard, the Regulation prevails [16]. According to the CPR, harmonised product standards were to be developed based on standardisation mandates issued by the Commission after consultation with a Standing Committee on Construction.

Another role was proposed for the second type of harmonised specifications, European Assessment Documents, which were supposed to replace ETAGs to provide the basis for assessing non-standardised products. The assessment of specific product performance is presented in ETA—European Technical Assessment (a successor of European Technical Approval) issued by the Technical Assessment Body (TAB), a member of EOTA from 2013 called the European Organisation for Technical Assessment. The process of EAD development is generally industry-driven as it starts when TAB receives an ETA application for product non-covered by the harmonised standard or for which assessment methods presented in the hEN do not recognise all product performances. This means that the ETA route reflects the needs of innovation, market novelties and products, for which standardisation might not be possible.

The review process of the CPR started at the end of 2019. Despite the comprehensive results of several studies completed in the period 2018–2019 (see EC website [17]), no clear picture has been indicted by the Commission of further development of the European legislation in the area of construction products. Within the studies, as member states, external parties were asked to provide a specific view on the range and content of the legislation, specific product related views, preferred approach to environmental issues, development of harmonised specifications, roles and responsibilities of different actors, etc.

Finally, more than seven years after implementing the current legislation, EU Regulation No 305/2011 [12], the European Commission presents potential options [18] for the review, that according to the growing interest of the EU Member States [19], should implement environmental sustainability principles and facilitate Circular Economy plan [20].

Later in the article mentioned past and current European legislation was evaluated on how it supported the free trade of construction products, especially in relation implementing sustainability principles. In the light of this evaluation, future options for the development of this legislation [18] were analysed and assessed to provide substantive conclusions that could be helpful for policymakers.

## 2. Materials and Methods

Qualitative research of the past and existing regulatory rules for construction products, based on the literature review (official EU documents, as regulations, directives, guidance, European Council conclusions, judgements, reports or studies), in the light of author's professional experience that was carried out. The main points of interest were: General progress of EU harmonisation rules, developments in standardisation, assessment of innovation and impact on the potential of implementation of sustainability principles. Similar assessment rules were applied to future policy options as presented by the European Commission.

In the discussion of results, an analysis was carried out, based on the summary of options, to identify the optimal approach to further development of construction products policy that would include environmental sustainability principles. This considered factors like relevant administrative impact on the industry and the Member States or potential to unlock progress of technical specifications that could include sustainability aspects and innovation-related tools.

## 3. Results

In this part of the article, considerations on how the evolution of the European approach to harmonisation of construction products affected or could affect the future implementation of environmental sustainability principles are presented.

In relation to the CPD, it needs to be assessed that the directive's very ambitious and optimistic objective, which was to harmonise national technical regulations on construction products, was never fully reached. In general, several deficiencies of its implementation and transposition were identified, from minor issues up to very serious ones. For example, according to the study [10] published in 2007, even such a basic issue, which was the meaning of the CE marking of construction product, was not fully and correctly recognised on the market. The CE mark for construction product was wrongly perceived as a safety or quality mark not related to the performances and its relation to use in construction works. As it was not mandatory in some Member States to CE mark construction products, the CE marking was naturally put in a weak position when the competition took place with national marks that had traditionally broad acceptance on domestic markets.

However, the main problem of implementation was linked to the fact that standardisation works (development of hENs) based on mandates were not responding in an effective way to the demands of industry, product development and evolution and the needs of the Member States new legal requirements. However, at the end of the CPD era, over 400 references to harmonised standards were published in OJEU. The alternative route of ETA was clearly filling the gap in the harmonised area to some extent, but the works of approval bodies still were slowed down by the mandating procedures. As a result, 35 ETAGs based on mandates were published by EOTA, followed by over 4300 ETAs (incl. issued based on art. 9(2)) [21].

This relatively slow mandating and standardisation process clearly affected the progress, especially of ER 3 implementation, because several Member States notified relevant national regulations, for example, emissions from construction products. On the other hand, horizontal standardisation works on complex testing methods for the release of dangerous substances based on mandate M/366 [22] and guidance [9] were also not fully completed and could not be included in relevant hENs. With the lack of complete harmonised solutions, the industry and some of the Member States developed and implemented national assessment procedures as supplement or alternative, for example, France [23] or Germany [24].

When the CPD was repealed and replaced with the CPR in 2013, great expectations came with the extended or new BWRs and potentially improved implementation principles. As CPR requirements are theoretically directly applicable in the same way in all Member States, there was a hope that at least some obstacles connected usually with national transposition could be avoided. This was not exactly the case, and now, with the perspective of the 2nd quarter of 2021, both legal and technical development under the Regulation must be assessed as only partly positive.

Officially, in the implementation report of 2016, the Commission [25] confirmed that national authorities and other stakeholders had already made legislative and administrative efforts both to ensure proper implementation of the CPR. Practically in most of the Member States inclusion of CPR into the national legal system was executed at least through a direct reference to Regulation 305/2011 and its application for construction products covered by hEN or ETA issued for them (e.g., "products covered by hEN or covered by ETA must be placed on the market in accordance with the Regulation (EU) No 305/2011") [26,27].

Therefore, it needs to be underlined that, considering the obligatory character of hEN implied by art. 4 of the regulation [28,29], the standardisation needs to be perceived as a core tool for European harmonisation of construction products under the CPR. Therefore, European Committee for Standardisation (CEN) developments are perceived as essential for the implementation process of the CPR, including environmentally-related BWRs 3, 6 and 7. Due to the fact that the methods for assessment of performance and classifications included in the standards harmonised with the CPD remained valid, they were all incorpo-

rated in the new system as standards harmonised with CPR on 1 July 2013. So, in relation to the environmental issues, they clearly represented a state of the art from the previous regime. Meanwhile, in 2012 the standardisation system and its procedures were affected by implementing the EU Standardisation Regulation [30]. Despite several attempts, starting from the application of the CPR, it needs to be underlined that no new mandate/request or amendment of the mandate was adopted and published by the Commission by mid-2021. With the last publication of new citations in March 2019 whole portfolio of harmonised standards under CPR covers c.a. 450 items, with around 420 dated back in the CPD era.

Regarding strictly environmental issues, from the formal point of view, consequently due to lack of mandating and standardisation progress, the potential of new BWR 7 and amended BWR 3 was still not achieved, even in comparison with the situation from the CPD era. Agreement was not reached on implementing environmental sustainability in the mandates, giving CEN no possibility to draft new hENs with BWR 7 and BWR 3 related clauses. A very similar lack of agreement concerns the establishment of specific AVCP systems for mentioned BWRs. For most cases, system "3" is suggested; however, this approach seems unsatisfactory for some stakeholders.

On the other hand, horizontal standardisation works are taking place based on Mandate M/350 [31], where CEN TC 350, established in 2005 (well before CPR and BWR 7), develops assessment methods of sustainability aspects of buildings and construction works. For construction products, the most relevant result of those works is EN 15804, which provides core product category rules for environmental declarations (EPDs). Meanwhile, the EPDs are widely used voluntarily, accepted and to some extent practically harmonised in the construction sector (see ECO-PLATFORM [32] development as an example). However, the Commission still analyses possibilities to use its own Product Environmental Footprint concept [33,34] for the CPR-related purposes. This conflicting approach (business to business vs business to consumer) seems to additionally slow down the whole process.

The general progress of non-standardised and innovative products seems to be far more visible. Despite several deficiencies claimed by the European Commission in the report [35], the overall picture for the EOTA route and the organisation itself must be perceived as positive. As a complement to main European standardisation, the ETA route offers a relatively quick solution for individual products, for which product standards based on wider consensus and fully standardised assessment and verification methods are not and will not be available in a predictable future. From 1 July 2013 to the end of 2020, over 8500 ETAs were issued by TABs. Subsequent development of harmonised specifications (EADs) resulted in the publication of 277 references; however, the number of adopted EADs available for issuance of ETA was much higher and exceeds 500. However, the works of EOTA covered products that are recycled [36], intended to be reused [37] or made from recycled materials [38], specific clauses on sustainable use of natural resources (BWR 7) that were developed by the Commission and EOTA are still not implemented in any of published EADs. Considering that, in general, the ETA route supports the effective growth of the innovation [39], the potential for practical implementation of BWR 7 seems to be still considerably high on EOTA's side.

This limited progress, especially in the field of publication of new references to harmonised standards, urged the Commission to open the discussion on the CPR review.

Consequently, in April 2020, the European Commission published the discussion document [18] on possible refined indicative options for the CPR review that presented several legislative options, starting from baseline scenario (no amendment) called option A, through different revision options (B to D) to repeal of the CPR (option E).

What could be their potential for achieving the general goals of the CPR and in relation to environmental sustainability aspects?

- The first option (A) is based on the general assumption that the increase of implementation effectiveness could be achieved via further use of informative guidance or soft law measures covering specific points of the regulation, without amendment of its main text. This action could concentrate mainly on specific market-relevant

issues, as the necessary content of the declaration of performance and CE marking, use of derogations described in art. 5 and simplified procedures for SMEs. As the standardisation appears to be the critical point for the Commission, this option covers improved cooperation with CEN and the comprehensive execution of the Action 5 of the Joint Initiative on Standardisation (JIS) [40] that should result in the publication of higher quality harmonised standards. Option A also implies further improvement of the performance of EOTA. However, due to a rather positive assessment of this performance expressed in preliminary results of the enquiry [41], such action could be perceived more as a business, a usual scenario, than a revolutionary one. Therefore, practically it does not bring additional administrative loads in comparison to the current situation. Regarding further inclusion of environmental aspects of construction products, the baseline scenario does not provide new fields of improvement. However, the main blockage of implementation of BWR 7 and BWR 3, together with the AVCP decisions, could be easily lifted when a significant backlog in the standardisation area is solved. As stated before, the EOTA route seems to be generally open and ready to take fully on board mentioned BWRs.

- More comprehensive actions concentrated on potential complex repair of the regulation are presented as the second review option (B). Under this scenario, implementation of the current CPR would be practically stopped for years as all the Commission's human resources would be involved in the review process [18]. Such revision could concentrate on coherence with the other pieces of EU legislation, an improvement on the route of development and potential adoption of harmonised specifications and specific measures necessary to implement environmental and circular economy principles. For innovation, the EC proposes a kind of temporary and preliminary CE marking based on draft harmonised specifications. What is most important, this option includes another distribution of the task and responsibilities in developing and adopting harmonised technical specifications. The whole responsibility would remain in the hands of the legislator (EC). The EC would also be empowered to precise additionally the area of technical harmonisation by delegated acts. According to the EC, this would open the possibility to the Member States to set additional technical requirements, including sustainability-related ones, without being in potential contradiction with the exhaustiveness principle. Under option B Commission also presents specific tools exclusively dedicated to environmental issues, such as Environmental Verification Organisations that could potentially take over some tasks from Notified Bodies. The most important is complete "repair of the CPR" using tools of option B would be a long-term process that would strongly involve the EC, national authorities, the industry and the other stakeholders. For example, developing completely new CPR Acquis (including technical specifications and supplementing acts) based on a current one could take between five to ten years [18].

- The third option (C) concerns partial focusing of the CPR to improve its coherence and versatility, using mechanisms for development and adoption of harmonised specifications as presented under option B. The harmonised language would be limited strictly to the performance assessment methods applicable to defined products and intended uses. Such an approach would exclude the common expression of performance and single specification per product type principle used in the construction world for decades. Therefore, the scope of the harmonisation could be restricted to groups of products relevant from the point of view of technical regulations of the Member States, environment and EU internal market. No specific improvement can be identified as regards the implementation of environmental or sustainability principles.

- Under option D, widening of the scope of the CPR is considered to include additionally inherent safety, health and environmental aspects of products (not included or not completely included in the current hENs). Therefore, sub-option D1 considers that product-related requirements could be applied more in line with the "New Approach" philosophy. In such a case, essential requirements would be formulated in the

standardisation request, based on which CEN would develop voluntary harmonised standards. The use of voluntary standards would give a non-exclusive assumption of conformity with the requirements. Sub-option D2 includes the possibility that specific product-related requirements would be covered in harmonised technical specifications developed according to option B's assumptions. Such a solution would be closer to the assumptions of the "Old Approach" philosophy.

- Finally, the Commission announced an option (E) with a possible repeal of the Regulation, with no replacement. This approach would totally switch the construction product market from current EU legislation to the EU mutual recognition principle. Taking into account that in such a case, member states would be given again the freedom to regulate without the use of common technical language, renaissance of national systems, national environmental and sustainability assessment schemes, and potential trade barriers could be expected.

In the light of this comparison, a summary of indicative options with regard to their potential impact (low, medium, high) on the future scope of EU harmonisation, implementation of innovation, implementation of sustainability and related administrative load (compared to the current one) is presented below in Table 1.

**Table 1.** Comparison of indicative options for the CPR review.

| Option | Harmonisation | Innovation | Sustainability | Administrative |
|:---:|:---:|:---:|:---:|:---:|
| A | medium | medium | medium | low |
| B | medium | low | medium | high |
| C | low | low | medium | high |
| D | high | low | high | high |
| E | low | low | low | high |

## 4. Discussion

Assessment of publicly available indicative options of the CPR review, in the light of the impact on the construction industry, improvement of general goals of harmonisation, implementation of innovation and environmental sustainability principles, shows many difficulties with a potential determination of optimal proposal for the future legislation.

It might be initially understood that relatively low future administrative load for the industry and the national authorities could be only linked to the baseline scenario described as option A, as it concentrates mainly on improving a standardisation process. What needs to be noted, under this option, the EC could still use existing powers to modify current CPR using delegated acts concerning, e.g., development of a specific AVCP system for environmental sustainability or horizontal notification for BWR 7 related characteristics. On the other hand, to improve implementation effectiveness, more intensive cooperation EC—CEN is needed. This covers both clear guidance from the side of the EC and modification on the side of CEN procedures. In the case of the second player in technical specifications and organisation responsible for technical assessments (EOTA), some improvement can be achieved in further cooperation with the EC by amendment of EAD development procedure, which is possible when the delegated act changes Annex II.

More radical options presented by the Commission, even if they could be perceived as new and revealing, in any case, would change the dramatically current paradigm of the European harmonisation and the meaning of relevant technical language. First of all, this is concerning the role of European institutions in the development of harmonised specifications. Taking into account that existing specification providers (CEN, EOTA) traditionally bring together and coordinate high-level practitioners and technical experts of the science and industry, it could be questionable if only a change of responsible party (to the EC itself) could improve the quality of the harmonised specifications, including proper, quicker or even more holistic implementation of environmental sustainability [42,43].

Considering the sole responsibility of Member States to regulate on a level of construction works, it seems not very feasible to change even partly legislative framework to "Old Approach" scheme. In such a case, detailed product requirements would be totally governed by the EU legislation, which would practically exclude variations on a national level. Therefore, wider harmonisation potentially could be applied to selected inherent product factors, as (user-related) safety. However, considering practical experiences of the construction sector, including the growth of staff professional qualifications, such extension may not be realistically needed for a wider range of products and applications.

As progress in the field of environmental sustainability needs to be linked directly with product, and even more general, industrial innovation, the future legislative framework should include dedicated tools that could directly and quickly respond to those challenges. Such dedicated tools should also facilitate innovation's need to shorten time-to-market and maintain the highest confidentiality standards when relevant. As market value needs to be considered, any kind of temporariness may negatively influence the perception of such a tool. Therefore, the existing concept of the EAD and ETA for non-standardised products needs to be appreciated, maintained, or even further developed (for example, improvement of time efficiency, transparency or SME's compatibility) to achieve a better response to EU sustainability and innovation principles.

With no replacement, the possible repeal of the regulation would totally switch the internal market from current EU legislation to the EU mutual recognition principle. In such a case, Union's potential to implement sustainability and a circular economy with coherent measures would be irrevocably delayed or even lost.

## 5. Conclusions

- This study shows high potential and relevance of continuation of the current regulatory framework for marketing of construction products. However, to unlock the capability of environmental sustainability, increased standardisation activity is needed in conjunction with smooth cooperation with the legislator. For support of innovation, no structural change is required as further development can be achieved with the BAU approach.
- The results of the research can be used in the process of policy formation or evaluation both on a national and international level. It provides qualitative analysis of the legislation (past, current) and publicly available policy revision options to identify future points of legal development that could connect low burden for the industry and national administrations with the general improvement of European harmonisation and with progress for sustainability and innovation in the construction sector.
- This study was based on literature research and on the author's own professional experience. No additional public consultation of the findings was carried out.
- The article considers the situation of the end of April 2021. Further studies will be conducted when the draft of the new Construction Product Regulation is published, likely at the beginning of 2022.

**Funding:** This research received no external funding.

**Institutional Review Board Statement:** Not applicable.

**Informed Consent Statement:** Not applicable.

**Data Availability Statement:** Not applicable.

**Conflicts of Interest:** The author declares no conflict of interest.

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
