# Peer review of "CE Marking of Construction Products—Evolution of the European Approach to Harmonisation of Construction Products in the Light of Environmental Sustainability Aspects"

_sustainability, doi:10.3390/su13116396_

Round 1

Reviewer 1 Report

Research issues are very current and in demand for practice. However, the research results are based only on the author's qualitative research and experience. Therefore, despite professional experience in this field, it is appropriate to improve this research in methodology. The following adjustments are therefore proposed:
1. In the abstract, it is necessary to describe how the research is processed in methodology or to specify the research goal in more detail.
2. The elaboration of the topic is based only on qualitative research and experience. Concerning measurability and objectivity, it is also necessary to incorporate quantitative methods based on the results.
3. The discussion part lacks more evident processing of the results (for example, in tabular form, specific proposals, etc.).

Author Response

Dear Sir/Madam,

first of all I would like to thank you for your comprehensive review of the article. To me it was extremely important to receive external feedback in order to change my personal perspective and approach to present, or what is even more relevant, to future research.

Following all comments received, generally I restructured the whole paper in order make it a bit more readable, therefore part of literature review was moved to the introduction, the content of short materials and methods section concentrates now on the nature of the analysis, and finally results and discussion should give clear picture of the scope of my research. I also added short conclusions to show final outcome, boundary conditions and the next steps.

Please find below my answers to your specific comments, which I tried to fully take into account:

  1. Clear statements on the methodology and aim of the research are now included in the abstract and in the other parts of the paper.
  2. Generally my research is currently based mainly on qualitative approach due to the nature of my professional work. However, following the advice I will try to implement quantitative measures in my further research. I suppose that expected publication of draft of the new CPR in more or less one year time will open new opportunities for such analysis.
  3. I tried to improve this part with a short table following results section which is followed by the discussion. As this table includes comparison of options with regard to certain criteria it should now be easier for reader to go through the findings. To complete the whole picture I added short conclusions section to show the outcomes, limitations and next steps.

Reviewer 2 Report

The paper provides an interesting read on "Evolution of the European Approach to Harmonisation of Construction Products in the Light of Environmental Sustainability 3 Aspects". 

However, there are some main points to revise before it is published.

1) There is no review of focal literature in this article. The introduction is not enough to pass as a literature review. It seems from the structure that materials and methods are the literature review. This is not very clear. If you are using materials and methods as your review of literature then you have to state it in the article.

2) The aim or purpose of the paper was not stated in the introduction. The author must succinctly provide a clear aim or objective for this article. 

3) The structure of the paper must be improved. I had to read it twice to understand the direction of the article. The structure should clearly state the aim, literature review, materials and methods.

4) You need to provide implications of findings after the discussion. Provide a clear discussion on the implications for policy formation, research and practice? 

5) The conclusion section must be provided in this article to elucidate limitations and further studies. 

Author Response

Dear Sir/Madam,

first of all I would like to thank you for your comprehensive review of the article. To me it was extremely important to receive external feedback in order to change my personal perspective and approach to present, or what is even more relevant, to future research.

Following all comments received, generally I restructured the whole paper in order make it a bit more readable, therefore part of literature review was moved to the introduction, the content of short materials and methods section concentrates now on the nature of the analysis, and finally results and discussion should give clear picture of the scope of my research. I also added short conclusions to show final outcome, boundary conditions and the next steps.

Please find below my answers to your specific comments, which I tried to fully take into account:

  1. The part of the (amended) literature review is now moved to the introduction part. Generally, in the materials and methods clear statement is now included that the qualitative research is based on literature review in conjunction with author’s professional experience.
  2. The scope of the paper, which is formation of background for policy makers on national and international level, is now clearly stated in the article.
  3. Following your advice, the structure of the article was amended in order to show clearly the direction of analysis.
  4. After discussion part I added short conclusions section in order to show clear results of the study, limitations and nest steps for my research.

Reviewer 3 Report

Dear Author,

Thank you for the manuscript with ID sustainability-1217127, the title "Evolution of the European approach to harmonisation of construction products in the light of environmental sustainability aspects". The manuscript is interesting, structured, and presented a new concept. Several comments regarding the improvement of this manuscript are as follow:

  1. The title of the manuscript must be represented the CE marking of a construction product; Please correct the title (please fill in the information regarding analysed problem of the topic: CE marking of a construction product);
  2. The literature review analysis with a demonstrated Essential Requirements ( ER1-6) can be presented in the manuscript;
  3. The literature review part with a analysis (with a research article, newly literature source) and literature reference list must be presented in the manuscript (in the review part the presented documents are welcome); Please correct the literature review part in the manuscript;
  4. The manuscript can be presented as a review article.

Reviewer

Author Response

Dear Sir/Madam,

first of all I would like to thank you for your comprehensive review of the article. To me it was extremely important to receive external feedback in order to change my personal perspective and approach to present, or what is even more relevant, to future research.

Following all comments received, generally I restructured the whole paper in order make it a bit more readable, therefore part of literature review was moved to the introduction, the content of short materials and methods section concentrates now on the nature of the analysis, and finally results and discussion should give clear picture of the scope of my research. I also added short conclusions to show final outcome, boundary conditions and the next steps.

Please find below my answers to your specific comments, which I tried to fully take into account:

  1. The title was changed accordingly.
  2. I added some basic documents concerning the whole package of ERs, however the paper clearly concentrates on ER - BWR 3 and BWR 7.
  3. I supplemented the literature review with some high relevant research articles in order to provide proper background for my analysis.
  4. This article is mainly based on the literature review, but in strong conjunction with author’s professional experience. In my opinion it goes so much beyond classical review article as it provides analysis not only of literature sources but also of the general situation of the construction products industry.

Round 2

Reviewer 1 Report

The post has been improved. Several comments were Incorporated

Reviewer 3 Report

Dear Authors,

Thank you for the corrections. The newly literature source in the manuscript (from 2020-2021 year) can be presented in the manuscript parts: literature analysis and reference list.

Reviewer

Author Response

Dear Sir/Madam,

Once again I would like to thank you for your review of the article.

Following your proposal I supplemented my article with literature references from 2020-2021.